# Hybrides of Alkaloid Lappaconitine with Pyrimidine Motif on the Anthranilic Acid Moiety: Design, Synthesis, and Investigation of Antinociceptive Potency

**DOI:** 10.3390/molecules25235578

**Published:** 2020-11-27

**Authors:** Kirill P. Cheremnykh, Victor A. Savelyev, Sergey A. Borisov, Igor D. Ivanov, Dmitry S. Baev, Tatyana G. Tolstikova, Valentin A. Vavilin, Elvira E. Shults

**Affiliations:** 1Novosibirsk Institute of Organic Chemistry, Siberian Branch, Russian Academy of Sciences, Lavrentjev Avenue 9, 630090 Novosibirsk, Russia; cherem@nioch.nsc.ru (K.P.C.); vicsav@nioch.nsc.ru (V.A.S.); sergalborisov@mail.ru (S.A.B.); mitja2001@gmail.com (D.S.B.); tolstiktg@nioch.nsc.ru (T.G.T.); 2The Federal Research Center Insitute of Molecular Biology and Biophysics, 2/12, Timakov St., 630117 Novosibirsk, Russia; diadoryh@ngs.ru (I.D.I.); drugsmet@niimbb.ru (V.A.V.)

**Keywords:** cross-coupling reaction, multi-component synthesis, alkaloids, lappaconitine, pyrimidine, analgesic activity

## Abstract

Convenient and efficient routes to construct hybrid molecules containing diterpene alkaloid lappaconitine and pyrimidine fragments are reported. One route takes place via first converting of lappaconitine to 1-ethynyl-lappaconitine, followed by the Sonogashira cross-coupling-cyclocondensation sequences. The other involves the palladium-catalyzed carbonylative Sonogashira reaction of 5′-iodolappaconitine with aryl acetylene and Mo (CO)_6_ as the CO source in acetonitrile and subsequent cyclocondensation reaction of the generated alkynone with amidines. The reaction proceeded cleanly in the presence of the PdCl_2_-(1-Ad)_2_PBn∙HBr catalytic system. The protocol provides mild reaction conditions, high yields, and high atom and step-economy. Pharmacological screening of lappaconitine-pyrimidine hybrids for antinociceptive activity in vivo revealed that these compounds possessed high activity in experimental pain models, which was dependent on the nature of the substituent in the 2 and 6 positions of the pyrimidine nucleus. Docking studies were undertaken to gain insight into the possible binding mode of these compounds with the voltage-gated sodium channel 1.7. The moderate toxicity of the leading compound **12** (50% lethal dose (LD_50_) value was more than 600 mg/kg in vivo) and cytotoxicity to cancer cell lines in vitro encouraged the further design of therapeutically relevant analogues based on this novel type of lappaconitine–pyrimidine hybrids.

## 1. Introduction

Diterpene alkaloid lappaconitine **1** is an important alkaloid of the aconite family [1,2]. It has been used clinically for more than 30 years in China to treat mild or moderate acute and chronic pains, including cancer-related pain, postoperative pain, and neuropathic pain [2,3,4]. In systemic animal studies, lappaconitine **1** has been reported to have properties of analgesia, anti-inflammation, local anesthesia, and hypothermia [3,4,5,6]. The analgesic activities of lappaconitine **1** are, generally, approximately seven times greater than those of phenazone, a commonly used non-steroidal anti-inflammatory drug, and are almost equipotent to pethidine and tramadol [2,7]. Moreover, multiple daily intraperitoneal injection of lappaconitine blocked spontaneous pain in a mouse leukemia bone cancer pain model [8]. The analgesic activity of lappaconitine **1** was not associated with the opioid system, as revealed by the findings that the actions were not antagonized by naloxone [9]. Moreover, clinical applications have demonstrated that the combination of lappaconitine and opioids produces a significant reduction in effective dosage and an enhancement in analgesic efficacy [10].

The mechanisms underlying lappaconitine analgesia are complex and debatable. It was reported that activation of the central noradrenergic and serotonergic systems might be involved in the antinociceptive effects. A reduction in the concentration of noradrenaline and damage to the serotonergic neurons in the central nervous system suppressed the analgesic activity of lappaconitine [11,12]. Another study argued that lappaconitine **1** may reduce the expression of the P2X3 receptor in rat dorsal root ganglionic neurons [13]. It was recently reported that various aconitum alkaloids, including lappaconitine **1** produced antinociception in a variety of rat models of chronic pain by stimulating spinal microglial dynorphin A release [14]. The in vitro data indicate that diterpene alkaloids can be grouped in Na^+^ channel activators and blockers [15]. According to [4,5], alkaloids that activate voltage-dependent Na^+^ channels are antinociceptive since they depolarize neurons permanently, and hence block the neuronal conduction. It is also found, that Na^+^ channel blockers possess antinociceptive activity by inhibiting neuronal activity [15]. Inhibition of neuronal activity and anti-epileptiform activity of lappaconitine **1** has been demonstrated by Ameri et al., on rat hippocampal slices in vitro [1]. This anti-epileptiform activity of lappaconitine **1** is also in line with their blockade of the Na^+^ channels since Na^+^ channels are known to be involved in the genesis of abnormal activities in epilepsy. Lappaconitine **1** frequency-dependently inhibit experimentally induced epileptiform activity by sparing the normal neuronal activity [1]. Consistent with the observed suppression of neuronal excitability in brain slices, lappaconitine **1** exhibited inhibitory effects on depolarization-elicited sodium currents in rat hippocampal and trigeminal ganglion neurons [16,17]. The Nav channel family contains nine members, which are essential for electrogenesis in excitable cells and are key targets for analgesics, anesthetics, antiepileptics, and antiarrhythmics. Developing subtype selective pharmacons that only block the channel isoforms (Nav1.1.–Nav1.9) specifically involved in the disease to be treated is a widely used strategy in modern drug research. Among Nav channels, Nav1.7 and Nav1.8, followed by Nav1.9, are the isoforms more implicated in pain signaling [18]. Recently, lappaconitine **1** was found to be an inhibitor of the Nav1.7 channel [19], which has been proposed as a promising analgesic target predominantly distributed in the nervous system [20].

It was reported that lappaconitine **1** has certain toxicities and its oral 50% lethal doses (LD_50_) in mice and rats are 32.4 and 20 mg/kg, respectively [21].

It is well known that the toxicity may result from the parent compound or from its metabolites. The metabolism in animals and humans have been clearly studied for lappaconitine **1**. The results obtained provided basic information to help better understand the pharmacological and toxicological activities of LAP in humans indicated that the ester bond of lappaconitine is stable [22,23,24]. Moreover, 2’-*N*-deacetyllappaconitine and 16-*O*-demethyllappaconitine are the main metabolic products of lappaconitine. The findings indicated that lappaconitine was widely metabolized: *N*-demethylation, O-demethylation, hydrolysis, and hydroxylation were main metabolic pathways, but that dehydroxylation and ester-exchange reactions were not found [24].

At the same time, the availability of lappaconitine **1** [25] has determined and caused the interest in synthesis of its derivatives with reduced toxic properties, and enhanced drugability.

We previously described modifications of lappaconitine **1** at the aromatic fragment, which reduced its toxicity and enhanced anti-arrhythmic properties [26].

In the framework of the systemic study of the effect of different constituents in the aromatic moiety on the antinociceptive potency of lappaconitine, we synthesized pyrimidine-containing lappaconitine derivatives and evaluated their analgesic activity in vivo by standard experimental pain models. Aryl substituted pyrimidines are considered as important pharmacophores possessing a wide diversity of valuable biological properties, including analgesic activity [27,28,29,30,31]. This scaffold was found to be a ubiquitous motif in many biologically active compounds and around these, a variety of novel drug candidates with excellent pharmaceutical profile can be designed. In particular, interest was considered to substituted pyrimidines promising for the treatment of neurological diseases [32,33]. The therapeutic effect of a variety of 2,4,5- and 2,4,6-trisubstituted pyrimidines is mainly attributed to an antagonistic effect to the adenosine receptors A_1_ [34,35] and A_2A_ [36,37]. The findings regarding the A_2A_ receptor signaling in pain have been controversial, with studies supporting both pro- and antinociceptive roles.

The biological activities of the pyrimidine derivatives indicated the maneuverability and versatility, which offer the medicinal chemist a continued interest in the pyrimidine skeleton. Exploiting pharmacophore hybridization and in the connection of optimization of the lappaconitine structure (for reducing the toxicity and improving the analgesic activity), we combined the two pharmacophoric units in a single entity. In the framework of our studies dealing with the development of the synthesis of pyrimidinyl-lappaconitine hybrid compounds [38], we report herein the convenient multi component strategy route to those compounds from lappaconitine **1**. The antinociceptive activity of new compounds was also investigated and discussed.

## 2. Results and Discussion

### 2.1. Chemical Synthesis

Previously, we successfully synthesized lappaconitine ynones **5**, **6** by the employment of transition-metal-catalyzed cross-coupling reactions of 5′-ethynyllappaconitine **2** [39] with benzoic acid chlorides **3**, **4** under Sonogashira cross-coupling reaction (Scheme 1) [38]. We observed that compounds **5** could be almost quantitatively converted into 2,4,6-trisubstituted pyrimidines **7**, **8** by refluxing with acetamidine hydrochloride **9** or guanidine hydrocarbonate **10** in acetonitrile in the presence of Na_2_CO_3_ (2 equivalent) with the isolated yield 81–95%.

In this work we found, that cyclocondensation of alkynyl ketones **5** and **6** with the less nucleophilic benzamidine hydrochloride **11** proceeds successfully in *i*-propanol in the presence of Et_3_N with the formation of compounds **12**, **13** (yield 65–67%).

For providing a most powerful approach to pyrimidines by limited number of reaction steps, we assumed that two-step Pd-catalyzed Sonogashira coupling and subsequent condensation could be carried out in a one-pot manner (in multicomponent reaction conditions) [40]. By using the above catalytic system (Pd[(PPh_3_)_2_]Cl_2_, Ph_3_P, CuI, Et_3_N) and carrying out the reaction of alkyne **2** with 4-bromobenzoyl chloride **14** in benzene for 7 h (TLC-control), evaporation of the solvent and subsequent condensation of the crude alkynone with acetamidine hydrochloride **9** or guanidine hydrocarbonate **10** in MeCN in the presence of a base (Na_2_CO_3_ or Et_3_N), we were able to isolate pyrimidines **15**, **16** in the yield of 70 and 75% respectively (Scheme 2). Thus, the one-pot procedure from alkyne **2** was also successfully employed for the synthesis of pyrimidines, substituted by a lappaconitine moiety.

However, the stability of the respective acid chlorides is limited and a lack of functional tolerance is the main problem of this methodology [41,42]. Kobayashi [43] and Beller [44] show that palladium-catalyzed carbonylative Sonogashira reaction (reaction of aryl halides with alkynes in the presence of carbon monoxide) represent a viable alternative and efficient methodology for the synthesis of alkynones. Simplicity, greater efficiency, and atom economy with generation of molecular complexity and diversity in the one-pot transformation are some of the advantages of these reactions.

We studied the reaction of 5′-iodolappaconitine **17** [45] with phenyl acetylene **18** in the presence of carbon monoxide (CO) (as a one-carbon building block) under palladium catalysis. In order to avoid handling of gaseous CO, several methods employing a variety of CO sources have been developed (metal carbonyls, formic acid, carbon dioxide, etc.) [46]. In our transformations we used the highly versatile CO source-molybdenum hexacarbonyl Mo (CO)_6_ [47]. As the palladium catalytic system in the three-component reaction of 5′-iodolappaconitine **17** with phenyl acetylene **18**, we used the practical Beller’s ligand [di(1-adamantyl)benzyl phosphonium bromide] Ad_2_PBn·HBr and PdCl_2_ as the source of palladium. This copper-free system was previously employed in Sonogashira coupling [48,49]. The reaction in MeCN proceeds for 2 h (65 °C, TLC-control) with the formation of 5′-(1-oxo-3-phenylprop-2-in-1-yl)lappaconitine **19** with the isolated yield 78% (Scheme 3). So, we have developed a one-pot copper-free procedure for the synthesis of alkynyl ketone **19** regioisomeric to alkynyl ketones **5**, **6**. The condensation of alkynone **19** with acetamidine hydrochloride **9** or guanidine carbonate **10** proceeded smoothly by heating in MeCN under reflux in the presence of a base (Na_2_CO_3_ for **12** or Et_3_N for **13**) and led to the formation of the desired pyrimidines **20**, **21** containing the diterpenic alkaloid lappaconitine fragment at the position C-4 (Scheme 3). Using the Beller catalyst was optimal for this carbonylative cross-coupling reaction. We investigated the carbonylative Sonogashira reaction of compound **17** and phenyl acetylene **18** and Mo(CO)_6_ (2 equivalent) with a catalyst system consisting of [PdCl_2_ (2 mol%)-Ph_3_P (4 mol%)] in the presence of Et_3_N (3 equivalent). Unfortunately, only 62% of the desired carbonylative product **19** at 85% conversion was obtained; as a major side-product the known non-carbonylative Sonogashira coupling product **22 [45]** (yield 14%) was formed. In the case of employing Pd (OAc)_2_ (2 mol%) as a palladium source, Ph_3_P (4 mol%) as the ligand and Et_3_N (3 equivalent) of this multicomponent reaction the desired compound **20** was not isolated. The formation of competing Sonogashira couplings (**19** (minor) and **22**) along with other non-identified compounds was observed by the analysis of the reaction mixture from the first step (NMR-control).

Next, we questioned whether the power and efficiency of this designed reaction sequence could be further improved, by combining in situ formation of the alkynyl ketone **19**, and its cyclocondensation reactions with amidines; thus, make the catalytic process more economic and attractive. We investigated the scope of this one-step consecutive multi-component reaction. As shown in Scheme 3, 5′-iodolappaconitine **17** was involved in the reaction with phenylacetylene **18** (1.8 equivalent) in the presence of molybdenum hexacarbonyl Mo(CO)_6_ (1.8 equivalent), a palladium catalytic system PdCl_2_ (3.4 mol %)−(1-Ad)_2_PBn∙HBr (5 mol %), and Et_3_N (3 equivalent) in acetonitrile for 2 h (TLC-control), and subsequent condensation of the crude reaction mixture with amidinium salt **9** or **10** and Et_3_N afforded pyrimidines **20**, **21** (62–71% isolated yield from **17**). Therefore, the palladium-catalyzed carbonylative Sonogashira reaction of 5′-iodolappaconitine 17 with aryl acetylene and Mo(CO)_6_ as the CO source in acetonitrile and subsequent cyclocondensation reaction of the generated alkynone with amidines presented an efficient step-economy approach to lappaconitine–pyrimidine hybrids.

Using the main principles of the palladium catalysis [50], it is possible to assume this transformation of 5′-iodolappaconitine **17** is a multistep sequence of events that involves, an initial oxidative addition of **17** to the LPd^0^ species to give the corresponding arylpalladium (II) complex (Scheme 4). Subsequent formation of the benzoyl palladium complex takes place by CO insertion. Supported by Et_3_N an exchange of the iodide by phenyl acetylide occurs and the desired product is formed by reductive elimination. It is worth mentioning that CO migration and insertion may be considered to be reversible steps within this system.

The composition and structure of the synthesized compounds were confirmed by IR spectroscopy, ^1^H and ^13^C NMR spectroscopy, and mass spectrometry. The ^1^H and ^13^C NMR spectra (see Appendix A) of the synthesized compounds contain one set of characteristic signals of the diterpene skeleton of lappaconitine and the pyrimidine. The formation of alkynone **19** is confirmed by their IR spectra containing an intense absorption band at 2200 cm^−1^ (stretching vibrations of the internal acetylene bond) and 1687, 1710 cm^−1^ (carbonyl functions). The ^1^H and ^13^C NMR spectra of compounds **9**, **10**, **12**, **13**, **15**, **16**, **20**, **21** contain signals of protons and carbon atoms characteristic of a 2,4,6-trisubstituted pyrimidine ring. Therefore, we developed convenient synthetic routes to hybrid compounds with lappaconitine and pyrimidine structural fragments **9**, **10**, **12**, **13**, **15**, **16**, **20**, **21**. Next, we studied their analgesic activity.

### 2.2. Biological Study

#### 2.2.1. Analgesic Activity

The analgesic activity was studied by standard experimental pain models, namely, the acetic acid-induced writhing (0.75% acetic acid, 0.1 mL per 10 g, intraperitoneal) and the hot plate (thermal stimulation, T = 54 °C) tests [5,51]. Agents were administered 1 h before testing at doses of 25 and 5 mg/kg. Diclofenac sodium at a dose of 10.0 mg/kg was used as reference drug. Lappaconitine **1** was used at the dose of 5 mg/kg; at a dose of 25 mg/kg this compound turned out to be very toxic (the death rate of mice was 100%). This result confirms the previously obtained data for lappaconitine **1**: the toxicity (LD_50_ dose) of lappaconitine was 32.4 mg/kg [21]. Table 1 presents the analgesic activity data of pyrimidine-lappaconitine in the acetic acid-induced writhing test (oral administration).

As can be seen from the presented data, compounds **10**, **12**, **13**, and **15** exhibited significant analgesic activity in the acetic acid-induced writhing test, especially at the dose of 5 mg/kg, which was comparable and almost higher to that of lappaconitine **1**. The activity of compound **10**, **12**, **13**, and **15** at a dose of 5 mg/kg was also comparable to Diclofenac sodium administered in a dose of 10 mg/kg (Table 1). Therefore we studied the activity of the indicated active compounds after oral and intraperitoneal administration at a dose of 1 mg/kg after. The data presented in Table 2.

Compounds **10**, **12**, **13**, and **15** exhibited reliable analgesic activity at a dose 1 mg/kg. Among them, compound **15** was the most active. When administered intraperitoneally at a dose of 1 mg/kg compounds **10**, **12**, and **13** retained their effect (MPE 28–63%). Interestingly, the intraperitoneal dose of 1 mg/kg was not enough for compound **15** to show a significant activity, while its analgesic effect in both routes of administration at a dose 5 mg/kg was practically the same (MPE 70% (oral, Table 1) and 64% (intraperitoneal, Table 2).

The analgesic activity of the compounds was further examined using an experimental model of thermal pain—the hot plate (Table 3).

As can be seen from the presented data, compounds **10**, **12**, **13**, and **15** at an oral dose of 5 mg/kg are significantly increased the time that animals spent on the hot plate, reducing the pain response value by 31–38%. Lappaconitine and Diclofenac sodium also exhibited a comparable reliable statistically significant analgesic effect. All other derivatives (**9**, **16**, **20**, and **21**) were inactive in both tests.

Considering the fact, that the most active compounds, that showed a significant analgesic effect in both tests, were **10**, **12**, **13**, and **15** it is most likely that the nature of the substituent in the 2 and 6 positions of the pyrimidine ring has an important role in this class of hybrid compounds. These compounds combined a 2-(amino)- and 6-(4-fluorophenyl)-substituent (for **10**), 2-(4-nitrophenyl)- and 6-(4-fluorophenyl)-substituent (for **12**), 2-(4-nitrophenyl)- and 6-(4-methoxyphenyl)-substituent (for **13**) or 2-methyl- and 6-(4-bromophenyl)-(for **15**). It can be noted, that two of the selected compounds (**12** and **13**) contain a 4-nitrophenyl-substituent in the 2 position of the pyrimidine ring. Additionally, three compounds **10**, **12**, and **15** have a 4-fluorophenyl or 4-bromophenyl-substituent in their structure. Compounds with un-substituted phenyl ring in the 6 position (**20**, **21**) were inactive.

#### 2.2.2. Toxicity and Cytotoxicity Studies

In order to afford additional data about the influence of the substituent in pyrimidine ring on the properties of the compounds **12** and **15** we studied the comparative cytotoxicity on human cancer cell lines. The method of double staining with the fluorescent dyes Hoechst 33342 and propidium iodide (PI) was used for obtaining data on the viability of human cell lines HepG2 (hepatocellular carcinoma), Hek293 (fetal kidney), U937 (histiocytic lymphoma), and Hep2 (laryngeal carcinoma). The tumor cell viability was analyzed using the “In Cell Investigator” program on an IN Cell Analyzer 2200 in an automatic mode to determine living, dead and apoptotic cells in the entire population. The results (Figure 1A–H and Figure 2A–H) are presented as the percentage of cells from two independent experiments.

From the results presented in Figure 1 and Figure 2, it is evident that there is a difference in cytotoxicity of the tested compound **12** and **15**. Compound **12** was less toxic on all four tested cell lines. A remarkable increase in cytotoxicity was observed for compound **15**, especially against Hek293 and U937 tumor cell lines; the significant difference in the cell death was found (Figure 2C,H,D,G). Figure 2B,E shows that compound **15** induced apoptosis on Hek293 and U-937 cells in concentration 30 and 10 μM, respectively. The obtained data shown that the 6-bromophenyl substituent in the pyrimidine ring increased the cytotoxicity of the lappaconitine–pyrimidine hybrids.

Thus, the in vivo and in vitro experiments revealed that compound **12** with the 6-(4-fluorophenyl)- and 2-(4-nitrophenyl) substituent in the pyrimidine ring can be defined as a leading compound for further investigations.

The lappaconitine derivative **12** was tested for acute toxicity in white outbred mice using a single intragastric administration according to the Kerber method. Compound **12** appeared to be moderately toxic substance, and its LD_50_ exceeded the 600 mg/kg value (oral administration). So the synthesized derivative 12 exhibits 20 times less toxicity compared to lappaconitine **1** [21].

### 2.3. Molecular Modeling of a Possible Mechanism of Antagonistic Effect of Lappaconitine 1 and New Lappaconitine Derivatives 10, 12, 15, on the Voltage-Gated Sodium Channel 1.7

Inhibitors selective for peripherally expressed Nav1.7, Nav1.8, and Nav1.9 isoforms have been identified as potential analgesics for the treatment of pain [18,19,52]. XRD studies of the active site of antagonists of the voltage-gated sodium channel 1.7 revealed three main pockets in its structure: anion-binding, selectivity, and lipid-exposed (Figure 3).

The anion-binding pocket of the active site is rich in charged amino acid residues (ARG1602, ARG1608), which enter into electrostatic interactions and hydrogen bonds with the heteroatoms of the thiadiazole ring and the sulfonamide group of the GX-936 antagonist.

The selectivity pocket provides for the binding of antagonists of certain sodium channel isoforms and contains amino acid residues (TYR1537, TRP1538), point substitutions of which lead to a sharp decrease in the inhibitory effect of antagonists. Amino acid residue TYR1537 enters into stacking interaction with benzonitrile and pyrazole rings of GX-936. The aromatic system of TRP1358 restricts the selectivity pocket from the outer part of the active site, entering into stacking interactions with the pyrazole and trifluoromethylphenyl rings of the antagonist molecule. The side chains of the amino acids VAL1541 and MET1582 provide stabilization of the trifluoromethylphenyl and benzonitrile rings due to van der Waals (hydrophobic) interactions. Amino acid residue ASP1586 is apparently capable of interacting with the pi-system of the trifluoromethylphenyl ring through the formation of electrostatic interaction.

The lipid-exposed pocket is directly adjacent to the phospholipid bilayer through which the Nav1.7 passes and contains the trifluoromethylphenyl group of the antagonist. In this regard, the part of the antagonist molecule located in this pocket practically does not interact with the protein structures of the channel; however, it can interact with phospholipid molecules, and has little effect on the selectivity profile of the antagonist (Figure 3 and Figure 4B).

Docking studies in the active site of sodium channel 1.7 for lappaconitine **1** and compounds **10**, **12**, and **15** was undertaken. All interactions are shown in Figure 4C–F.

The structural peculiarity of compounds with in vivo activity **10**, **12**, and **15** was the pyrimidine core with lappaconitine residue in the 4 position and different substituent in 2 (amino-, 4-nitrophenyl- or methyl-) and 6 (4-fluorophenyl-, 4-bromophenyl-) positions. Inspection of the binding mode demonstrated, that the lappaconitine core of new hybrid compounds **10**, **12**, **15** remains outside the binding site of the Nav1.7 antagonists and is located on the outside of the protein molecule adjacent to the phospholipid bilayer. The substituents are unable to penetrate into the anion-binding pocket of the binding site, but they actively interact with the selectivity pocket, which may contribute to the development of the effect observed in in vivo experiments (Figure 4A).

Compounds **10** and **15** are characterized by the possibility of forming stacking interactions between the aromatic systems of their phenyl and pyrimidine rings and the π-system of amino acid residue TYR1537. The possible conformation of the compound **12** is very different in the location of the lappaconitine core, which leads to the formation of stacking between its phenyl, pyrimidine and 2-acetylamino-benzoate residue and the π-systems of the side chains of both amino acids TYR1537 and TRP1538 important for selectivity, as in the case of the GX-936 antagonist. The formation of electrostatic interactions between the π-systems of the pyrimidine and 2-acetylmino-benzoate moiety of compound **10** and the anions of amino acid residues ASP1586 and GLU1589 is possible. In the case of compounds **12** and **15**, such electrostatic interaction can occur only between the π-system of the pyrimidine ring and the ASP1586 anion. Compound **10** is characterized by the possible formation of a weak halogen interaction between the fluorine atom of the 4-fluorophenyl substituent and the MET1582 amino acid residue. In the case of compound **12**, such a bond possibly occurs between the fluorine atom of the 4-fluorophenyl ring and the side chain of amino acid GLN1530. These interactions can increase the degree of affinity of the compounds, causing a more pronounced effect in in vivo experiments compared to compounds containing an un-substituted phenyl ring (for example **20**, **21**). In addition, for compound **12**, there is a possibility of interaction between the π-system of the 2-acetylamino-benzoate residue and the sulfur atom of amino acid residue MET1582. An interesting feature of the **10** conformation at the binding site is the close position of the amino group of the 2-aminopyrimidine ring to one of the phospholipid molecules (PX41807). This can promote the formation of hydrogen bonds, which demonstrates the ability of molecules of this type to interact with the phospholipid bilayer, penetrating through it. The lappaconitine core of the **10** and **15** molecules is stabilized on the lateral surface of the Nav1.7 due to the formation of a hydrogen bond with the side chain of the amino acid GLN1528, which is far enough from the antagonist binding site. Due to the absence of substituent on the 2-acetylamino benzoate moiety, the lappaconitine molecule **1** can be located much closer to the lipid-exposed pocket of the binding site than the lappaconitine core of the hybrid compounds **10**, **12** and **15**. This position of the molecule **1** (Figure 4C) can lead to the formation of hydrogen bonds between the protons of the hydroxyl groups of the 8 and 9 position of the lappaconitine core and the acetylamino group of the lappaconitine anthranilate moiety and the carboxyl group of amino acid residue GLU1589.

The results of the docking experiments revealed the peculiarity of interactions of compounds **10**, **12**, **15** and lappaconitine **22** with the sodium channel 1.7 binding site.

## 3. Experimental Section

### 3.1. General Information

IR spectra were recorded on a Bruker Vector 22 FTIR spectrometer (Billerica, MA, USA) in KBr pellets. ^1^H and ^13^C NMR spectra were acquired on Bruker AV-400 (400.13 (^1^H), 100.76 (^13^C) MHz) or DRX-500 (500.13 (^1^H), 125.76 (^13^C) MHz) spectrometers (Billerica, MA, USA). Deuterochloroform (CDCl_3_) was used as a solvent, with residual CHCl_3_ (δ_H_ = 7.26 ppm) or CDCl_3_ (δ_C_ = 77.0 ppm) being employed as internal standards. Chemical shift was expressed in ppm (δ). NMR signal assignments were carried out with the aid of a combination of 1D and 2D NMR techniques that included Heteronuclear Single Quantum Correlation (HSQC) and Heteronuclear Multiple Bond Correlation (HMBC). Signals in the NMR ^1^H and ^13^C spectra of the lappaconitine part of new compounds were assigned by correlation with those of lappaconitine **1** [53]. Copies of NMR ^1^H and ^13^C spectra of all new compounds a given in Appendix A. The mass spectra were recorded on a Thermo Scientific DFS high-resolution mass spectrometer (evaporator temperature 200–250 °C, EI ionization at 70 eV). Melting points were determined using thermo system Mettler Toledo FP900 (Columbus, OH, USA).

The reaction progress was monitored by TLC on Silufol UV-254 plates (Kavalier, Czech Republic, CHCl_3_-EtOH, 100:1; detection under UV light or by spraying the plates with 10% water solution of H_2_SO_4_ followed by heating at 100 °C). Preparative column chromatography was carried out on 60H silica gel (0.063–0.200 mm, Merck KGaA, Darmstadt, Germany) with the indicated solvent systems. Oxygen or water sensitive reactions were performed under the argon atmosphere. The starting material: 5′-ethynyllappaconitine **2** [39], 5′-iodolappaconitine **17** [45] and Ad_2_PBn^·^HBr [54,55] were synthesized, according to previously published procedures. Other reagents were purchased from commercial sources and were used without further purification. Solvents (CH_3_CN, *i*-PrOH, benzene, CHCl_3_, CH_3_OH) and Et_3_N were purified by standard methods. Purity of all compounds was checked by TLC.

(8,9-Dihydroxy-1α,14α,16β-trimethoxy-20-ethylaconitan4β-yl)2-acetylamino-5-[3-(4-fluorophenyl)-3-oxoprop-1-yn1-yl]benzoate (**5**), (8,9-dihydroxy-1α,14α,16β-trimethoxy-20-ethylaconitan-4β-yl) 2-acetylamino-5-[3-(4-methoxyphenyl)-3-oxoprop-1-yn-1-yl]benzoate (**6**), (8,9-dihydroxy-1α,14α,16β-trimethoxy-20-ethylaconitan-4β-yl) 2-acetylamino-5-[6-(4-fluorophenyl)-2-methylpyrimidin-4-yl]benzoate (**9**), (8,9-dihydroxy-1α,14α,16β-trimethoxy-20-ethylaconitan-4β-yl) 5-[2-amino-6-(4-fluorophenyl)pyrimidin-4-yl]-2-acetylaminobenzoate (**10**) were prepared as reported before [38].

### 3.2. Syntheses and Spectral Data

#### 3.2.1. Carbonylative Sonogashira Reaction

To solution of Et_3_N (101 mg, 1 mmol) in CH_3_CN (4 mL) PdCl_2_ (2 mg, 0.011 mmol), and Ad_2_PBn·HBr (7 mg, 0.017 mmol) were successively added under a stream of argon with stirring. The mixture was heated to 65 °C (bath) and Mo(CO)_6_ (158 mg, 0.59 mmol), phenylacetylene **18** (60 mg, 0.59 mmol) and 5′-iodolappaconitine **17** (236 mg, 0.33 mmol) were added. The reaction mixture was heated until 2 h, could and the precipitate was filtered and washed with CH_3_CN (2 mL). The combined solvent was evaporated under reduced pressure. The residue was purified by column chromatography (benzene or CHCl_3_-EtOH, 50:1) to afford the corresponding compound **20** or **21**.

4β-(2′-acetylamino-5′-(7′-oxo-9′-phenylprop-8′-in-7′-yl)benzoate)-1α,14α,16β-trimethoxy-20-ethylaconitane-8,9-diol (**19**) (159 mg, yield 78%).

Yellowish solid, mp 118.2 (decomp.); ^1^H NMR (400 MHz, CDCl_3_, δ, ppm): 1.12 (t, 3H, *J* = 7.1 Hz, CH_3_-22), 1.55 (dd, 1H, *J* = 14.8, 8.3 Hz, H-6β), 1.86–2.65 (m, 16H, H-3β, H-12β, H-15β, H-10, H-2β, H-7, H-2α, CH-13, H-15α, H-5, H-12α, H-21α, H-19β, H-21β, OH, H-3α), 2.24 (s, 3H, CH_3_C(O)NH), 2.71 (dd, 1H, *J* = 14.8, 7.4 Hz, H-6α), 3.01 (s, 1H, H-17); 3.19 (dd, 1H, *J* = 10.2, 7.1 Hz, H-1), 3.27–3.33 (m, 1H, H-16), 3.29 (s, 3H, CH_3_O-1), 3.30 (s, 3H, CH_3_O-16), 3.40 (s, 3H, CH_3_O-14), 3.43 (d, 1H, *J* = 4.7 Hz, H-14), 3.51 (d, 1H, *J* = 11.4 Hz, H-19α), 3.52 (br.s, 1H, OH), 7.44 (m, 3H, H-3″,4″,5″); 7.68 (m, 2H, H-2″,6″), 8.31 (dd, 1H, *J* = 8.9, 2.1 Hz, H-4′), 8.74 (d, 1H, *J* = 2.1 Hz, H-6′); 8.84 (d, 1H, *J* = 8.9 Hz, H-3′), 11.40 (s, 1H, NH); ^13^C NMR (101 MHz, CDCl_3_, δ, ppm): 13.4 (CH_3_-22), 24.0 (C-6), 25.5 (CH_3_), 26.0 (C-12), 26.6 (C-2), 31.7 (C-3), 36.1 (C-13), 44.7 (C-15), 47.4 (C-7), 47.8 (C-5), 48.8 (C-21), 49.6 (C-10) 50.8 (C-11), 55.3 (C-19), 56.0 (CH_3_O-16), 56.4 (CH_3_O-1), 57.8 (CH_3_O-14), 61.4 (C-17), 75.4 (C-8), 78.4 (C-9), 82.7 (C-16), 83.9 (C-1), 85.7 (C-4), 86.5 (C-8′), 90.0 (C-14), 93.3 (C-7′), 115.2 (C-1′), 119.8 (C-3′), 120.0 (C-1″), 128.7 (C-3″, 5″), 130.6 (C-4″), 130.7 (C-5′), 133.0 (C-2″,6″), 133.0 (C-6′), 135.0 (C-4′), 146.2 (C-2′), 166.6 (C=O), 169.3 (C(O)NH), 175.9 (C=O); IR (KBr, ν, cm^−1^): 3458 (OH, NH), 2200 (C≡C), 1710 (C=O), 1687 (C=O amide), 1585 (C=N); HR-MS (ESI), calcd. for C_41_H_48_O_9_N_2_: 712.3354; found, [M]^+^
*m*/*z*: 712.3340.

#### 3.2.2. Procedures for Cross-Coupling Reactions

(a)A mixture of alkynyl ketone **5** or **6** (0.3 mmol), 4-nitrobenzamidine hydrochloride **11** (72.5 mg, 0.36 mmol) and Et_3_N (90.9 mg, 0.9 mmol) was reflux under stirring in *i*-PrOH (8 mL) 8 h (TLC). After cooling (about 10 h), the precipitate was filtered, washed with CHCl_3_ (10 mL), the combined organic layer was concentrated under reduced pressure, and the residue was subjected to column chromatography (chloroform-EtOH). Fraction with R_f_ 0.4 (chloroform-EtOH, 20:1), was evaporated and dried in vacuo to give compounds **12**, **13** as a yellow powders.(b)One-pot cross-coupling/cyclocondensation. To a stirred mixture of Pd(PPh_3_)_2_Cl_2_ (4.3 mg, 0.005 mmol), PPh_3_ (2.1 mg, 0.007 mmol), CuI (2.5 mg, 0.01 mmol) in benzene (5 mL) under argon flow was added Et_3_N (206 mg, 0.27 mL, 2 mmol) followed by freshly prepared 4-bromobenzoyl chloride **14** (132 mg, 0.6 mmol) in benzene (2 mL). The 5′-Ethynyllappaconitine **2** (304 mg, 0.5 mmol) in benzene (5 mL) was added dropwise during one hour. The reaction mixture was heated to 65 °C under stirring for 7 h (TLC-control). After removing the solvent in vacuo under argon, the crude material was dissolved in acetonitrile (7 mL), and acetamidine hydrochloride **9** (67 mg, 0.7 mmol) and anhydrous Na_2_CO_3_ (170 mg, 1.6 mmol), or guanidine hydrocarbonate **10** (41 mg, 0.7 mmol) and Et_3_N (162 mg, 1.6 mmol) were added. The reaction mixture was stirred under reflux for 8 h. After cooling (about 8 h) the precipitate was filtered, washed with CHCl_3_ (10 mL), the combined organic layer was washed with water (2 × 5 mL), dried over magnesium sulfate and filtered. The solvent was removed under reduced pressure, and the residue was subjected to column chromatography (chloroform-EtOH). The fraction with R_f_ 0.4 (chloroform-EtOH, 20:1), was collected evaporated and dried in vacuo to give compounds **15** or **16** as a yellow powders.(c)One-pot carbonylative cross-coupling/cyclocondensation. To solution of Et_3_N (101 mg, 1 mmol) in CH_3_CN (7 mL) PdCl_2_ (2 mg, 0.011 mmol), and Ad_2_PBn·HBr (7 mg, 0.017 mmol) were successively added under a stream of argon with stirring. The mixture was heated to 65 °C (bath) and Mo(CO)_6_ (158 mg, 0.59 mmol), phenylacetylene **18** (60 mg, 0.59 mmol) and 5′-iodolappaconitine **17** (236 mg, 0.33 mmol) were added. The reaction mixture was heated until 2 h at 65 °C, treated with amidinium salt **9** (0.46 mmol) and anhydrous Na_2_CO_3_ (106 mg, 1 mmol), or guanidine carbonate **10** (0.46 mmol) and Et_3_N (101 mg, 1 mmol) and stirred under reflux for 8 h. After completion based on TLC, the cooled mixture was filtered over Celite, the precipitate washed with CHCl_3_ (15 mL), the combined solvent was removed under reduced pressure, and the residue was subjected to column chromatography (chloroform-EtOH, 80:1). Fraction with R_f_ 0.4 (chloroform-EtOH, 20:1) was evaporated and dried in vacuo to give compounds **20**, **21** as yellow powders.

#### 3.2.3. 4β-{2′-Acetylamino-5′-[11′-(4-fluorophenyl)-9′-(4-nitrophenyl)-pyrimidine-7′-yl]benzoate}-1α,14α,16β-trimethoxy-20-ethylaconitane-8,9-diol (**12**)

Yield 67% (a); yellow amorphous powder; mp 169.8 °C (decomp.); ^1^H NMR (400 MHz, CDCl_3_, δ, ppm): 1.16 (t, 3H, J = 7.1 Hz, CH_3_-22), 1.70–1.84 (m, 2H, H-3β, H-6β), 1.95–2.09 (m, 2H, H-12β, H-15β), 2.12 (dd, 1H, J = 12.4, 4.4 Hz, H-10), 2.16–2.35 (m, 3H, H-2β, H-2α, H-7), 2.27 (s, 3H, CH_3_C(O)), 2.35–2.68 (m, 8H, H-12α, H-15α, H-21α, H-19β, H-21β, H-5, H-13, OH), 2.78 (dd, 1H, *J* = 12.5, 2.2 Hz, H-3α), 2.93 (dd, 1H, *J* = 14.8, 7.4 Hz, H-6α), 3.05 (s, 1H, H-17), 3.20 (dd, 1H, *J* = 10.2, 7.1 Hz, H-1), 3.26–3.35 (m, 1H, H-16), 3.30 (s, 3H, CH_3_O-1), 3.32 (s, 3H, CH_3_O-16), 3.43 (s, 3H, CH_3_O-14), 3.46 (d, 1H, *J* = 4.7 Hz, H-14), 3.48 (br.s, 1H, OH), 3.70 (d, 1H, *J* = 11.4 Hz, H-19α), 7.23 (t, 2H, *J* = 8.5 Hz, H-3″,5″), 7.94 (s, 1H, H-12′), 8.22–8.29 (m, 3H, H-2″,6″,4′), 8.48 (d, 2H, *J* = 8.7, H-2′′′,6′′′), 8.81 (d, 2H, *J* = 8.7, H-3′′′,5′′′), 8.88 (d, 1H, *J* = 8.8, H-3′), 9.03 (d, 1H, *J* = 2.1, H-6′), 11.25 (s, 1H, NH); ^13^C NMR (101 MHz, CDCl_3_, δ, ppm): 13.4 (CH_3_-22), 24.0 (C-6), 25.5 (CH_3_C(O)), 26.0 (C-12), 26.6 (C-2), 31.7 (C-3), 36.1 (C-13), 44.7 (C-15), 47.4 (C-7), 48.9 (C-5), 48.8 (C-21), 49.6 (C-10), 50.8 (C-11), 55.3 (C-19), 56.0 (CH_3_O-16), 56.4 (CH_3_O-1), 57.8 (CH_3_O-14), 61.4 (C-17), 75.4 (C-8), 78.4 (C-9), 82.7 (C-16), 83.9 (C-1), 85.0 (C-4), 89.9 (C-14), 109.3 (C-12′), 115.9 (d, C, ^2^*J*_CF_ = 21.1 Hz, C-3″,5″), 116.0 (C-1′), 120.2 (C-3′); 123.6 (C-3′′′,5′′′), 130.0 (C-2′′′,6′′′), 128.1 (d, ^3^*J*_CF_ = 8.8 Hz, C-2″,6″), 129.8 (C-5′), 130.2 (C-6′), 132.0 (C-4′), 132.7 (d, ^4^*J*_CF_ = 3.1 Hz, C-1″), 143.3 (C-2′), 143.7 (C-1′′′), 149.2 (C-4′′′), 162.0 (C-7′); 162.7 (C-11′), 163.7 (C-9′), 164.6 (d, *J*_CF_ = 252.2 Hz, C-4ʹʹ), 166.9 (C=O), 169.1 (CH3C(O)NH); IR (KBr, ν, cm^−1^): 3392 (OH, NH), 1703 (C=O), 1686 (amide C=O), 1578 (C=N), 1522 (ArNO2), 1342 (Ar-NO2), 1228 (C-F), 840, 760, 725 (C=C); HR-MS (ESI), *m*/*z* (I*_rel_*, %): 877 (1), 846 (3), 577 (2), 576 (2), 410 (6), 405 (6), 392 (15), 376 (100), 374 (8), 360 (5), 348 (5), 91 (2), 41 (2), calcd. for C_48_H_52_FN_5_O_10_: 877.4811; found, [M − OCH_3_]^+^
*m*/*z*: 846.3512.

#### 3.2.4. 4β-{2′-Acetylamino-5′-[11′-(4-methoxyphenyl)-9′-(4-nitrophenyl)-pyrimidine-7′-yl]benzoate}-1α,14α,16β-trimethoxy-20-ethylaconitane-8,9-diol (**13**)

Yield 64% (a); yellow amorphous powder; mp 152.2 °C (decomp.); ^1^H NMR (400 MHz, CDCl_3_, δ, ppm): 1.17 (t, 3H, *J* = 7.1 Hz, CH_3_-22), 1.69–1.83 (m, 2H, H-3β, H-6β), 1.94–2.14 (m, 3H, H-12β, H-15β, H-10), 2.16–2.25 (m, 2H, H-2β, H-7), 2.23 (s, 3H, CH_3_C(O)), 2.30–2.70 (m, 9H, H-2α, H-12α, H-15α, H-21α, H-19β, H-21β, H-5, H-13, OH), 2.77 (dd, 1H, *J* = 12.5, 2.4 Hz, H-3α), 2.93 (dd, 1H, *J* = 14.8, 7.4 Hz, H-6α), 3.04 (s, 1H, H-17), 3.19 (dd, 1H, *J* = 10.2, 7.1 Hz, H-1), 3.28–3.36 (m, 1H, H-16), 3.30 (s, 3H, CH_3_O-1), 3.32 (s, 3H, CH_3_O-16), 3.44 (s, 3H, CH_3_O-14), 3.44 (d, 1H, *J* = 4.8 Hz, H-14), 3.52 (br.s, 1H, OH), 3.68 (d, 1H, *J* = 11.4 Hz, H-19α), 3.88 (s, 3H, CH_3_O-4″), 6.97 (d, 2H, *J* = 8.6 Hz, H-3″,5″), 7.68 (s, 1H, H-12′), 8.04-8.14 (m, 3H, H-2″,6″,4′), 8.37 (d, 2H, *J* = 8.8, H-2′′′,6′′′), 8.64 (d, 2H, *J* = 8.8, H-3′′′,5′′′), 8.77 (d, 1H, *J* = 8.8, H-3′), 8.85 (d, 1H, *J* = 2.1, H-6′), 11.25 (s, 1H, NH); ^13^C NMR (101 MHz, CDCl_3_, δ, ppm): 13.4 (CH_3_-22), 24.0 (C-6), 25.5 (CH_3_C(O)), 26.0 (C-12), 26.6 (C-2), 31.7 (C-3), 36.1 (C-13), 44.7 (C-15), 47.4 (C-7), 48.9 (C-5), 48.8 (C-21), 49.6 (C-10), 50.8 (C-11), 55.2 (CH_3_O-4″), 55.3 (C-19), 56.0 (CH_3_O-16), 56.4 (CH_3_O-1), 57.8 (CH_3_O-14), 61.4 (C-17), 75.4 (C-8), 78.4 (C-9), 82.7 (C-16), 83.9 (C-1), 85.0 (C-4), 89.9 (C-14), 108.4 (C-12′), 114.1 (C-3″,5″), 115.7 (C-1′), 120.0 (C-3′); 123.5 (C-3′′′,5′′′), 128.5 (C-2″,6″), 128.7 (C-1″), 128.8 (C-2′′′,6′′′), 129.9 (C-5′), 130.0 (C-6′), 131.9 (C-4′), 143.5 (C-2′), 144.0 (C-1′′′), 148.9 (C-4′′′), 161.5 (C-7′); 161.9 (C-9′), 162.0 (C-4″), 163.9 (C-11′), 166.9 (C=O), 169.1 (CH_3_C(O)NH). IR (KBr, ν, cm^−1^): 3390 (OH, NH), 1705 (C=O), 1685 (amide C=O), 1578 (C=N), 1520 (ArNO_2_), 1340 (Ar-NO_2_), 839, 754 (C=C); HR-MS (ESI), *m*/*z* (I*_rel_*, %): 889 (0.5), 831 (1), 719 (1), 501 (15), 470 (6), 459 (5), 430 (100), 400 (23), 386 (21), 376 (35), 223 (6), calcd. for C_49_H_55_N_5_O_11_: 889.3898; found, [M − NHC(O)CH_3_]^+^
*m*/*z*: 831.3610.

#### 3.2.5. 4β-{2′-Acetylamino-5′-[11′-(4-bromophenyl)-9′-(methyl)pyrimidine-7′-yl]benzoate}-1α,14α,16β-trimethoxy-20-ethylaconitane-8,9-diol (**15**) 

Yield 70% (b); yellow amorphous powder; mp 153.1 °C (decomp.); ^1^H NMR (500 MHz, CDCl_3_, δ, ppm): 1.12 (t, 3H, *J* = 7.1 Hz, CH_3_-22), 1.66 (dd, 1H, *J* = 14.8, 8.3 Hz, H-6β), 1.85 (br.t, 1H, *J* = 12.5 Hz, H-3β), 1.91–1.96 (m, 1H, H-12β), 2.00–2.08 (m, 1H, H-15β), 2.10 (dd, 1H, *J* = 12.4, 4.4 Hz, H-10), 2.15–2.22 (m, 2H, H-2β, H-7), 2.25 (s, 3H, CH_3_C(O)), 2.26–2.33 (m, 2H, H-2α, H-13), 2.34–2.44 (m, 2H, H-15α, H-5), 2.45–2.54 (3H, m, H-12α, H-21α, H-19β), 2.58–2.68 (m, 2H, H-21β, OH), 2.70 (dd, 1H, *J* = 12.5, 2.4 Hz, H-3α), 2.78 (dd, 1H, *J* = 14.8, 7.4 Hz, H-6α), 2.81 (s, 3H, CH_3_-pyrimidine), 3.02 (s, 1H, H-17), 3.19 (dd, 1H, *J* = 10.2, 7.1 Hz, H-1), 3.27–3.33 (m, 1H, H-16), 3.29 (s, 3H, CH_3_O-1), 3.30 (s, 3H, CH_3_O-16), 3.40 (s, 3H, CH_3_O-14), 3.43 (d, 1H, *J* = 4.8 Hz, H-14), 3.53 (br.s, 1H, OH), 3.60 (d, 1H, *J* = 11.4 Hz, H-19α), 7.62 (d, 2H, *J* = 8.6 Hz, H-3″,5″), 7.76 (s, 1H, H-12′), 7.98 (d, 2H, *J* = 8.6 Hz, H-2″,6″), 8.22 (dd, 1H, *J* = 8.8, 2.2 Hz, H-4′), 8.72 (d, 1H, *J* = 2.2, H-6′), 8.82 (d, 1H, *J* = 8.8, H-3′), 11.11 (s, 1H, NH); ^13^C NMR (125 MHz, CDCl_3_, δ, ppm): 13.4 (CH_3_-22), 24.0 (C-6), 25.5 (CH_3_C(O)), 26.0 (C-12), 26.3 (CH_3_ pyrimidine), 26.6 (C-2), 31.7 (C-3), 36.1 (C-13), 44.7 (C-15), 47.4 (C-7), 48.4 (C-5), 48.8 (C-21), 49.6 (C-10), 50.8 (C-11), 55.3 (C-19), 56.0 (CH_3_O-16), 56.4 (CH_3_O-1), 57.8 (CH_3_O-14), 61.4 (C-17), 75.4 (C-8), 78.4 (C-9), 82.7 (C-16), 83.9 (C-1), 85.0 (C-4), 89.9 (C-14), 108.8 (C-12′), 116.0 (C-1′), 120.4 (C-3′); 125.2 (C-4″), 128.7 (C-2″,6″), 130.1 (C-6′), 130.9 (C-5′), 132.0 (C-3″,5″), 132.6 (C-4′), 136.2 (C-1″), 143.4 (C-2′), 163.2 (C-7′); 163.5 (C-11′), 166.9 (C=O), 168.6 (C-9′), 169.0 (CH_3_C(O)NH). IR (KBr, ν, cm^−1^): 3394 (OH, NH), 1705 (C=O), 1685 (C=O amide), 1537, 1585 (C=N); HR-MS (ESI), *m*/*z* (I*_rel_*, %): 830 (1), 799 (2), 577 (2), 576 (2), 456 (3), 427 (13), 425 (13), 410 (14), 409 (12), 405 (17), 392 (16), 377 (25), 376 (100), 374 (17), 43 (11); calcd. for C_43_H_51_BrN_4_O_8_: 830.2700; found, [M − OCH_3_]^+^
*m*/*z*: 799.2705.

#### 3.2.6. 4β-{2′-Acetylamino-5′-[9′-(amino)-11′-(4-bromophenyl)pyrimidine-7′-yl]benzoate}-1α,14α,16β-trimethoxy-20-ethylaconitane-8,9-diol (**16**)

Yield 75% (b); yellow amorphous powder; mp 164.8 °C (decomp.); ^1^H NMR (400 MHz, CDCl_3_, δ, ppm): 1.12 (t, 3H, *J* = 7.1 Hz, CH_3_-22), 1.67 (dd, 1H, *J* = 14.8, 8.3 Hz, H-6β), 1.84 (br.t, 1H, *J* = 12.5 Hz, H-3β), 1.91–1.99 (m, 1H, H-12β), 2.04–2.09 (m, 1H, H-15β), 2.10 (dd, 1H, *J* = 12.4, 4.4 Hz, H-10), 2.15–2.22 (m, 2H, H-2β, H-7), 2.25 (s, 3H, CH_3_C(O)), 2.24–2.33 (m, 2H, H-2α, H-13), 2.34–2.44 (m, 2H, H-15α, H-5), 2.44–2.54 (3H, m, H-12α, H-21α, H-19β), 2.55–2.64 (m, 2H, H-21β, OH), 2.69 (dd, 1H, *J* = 12.5, 2.4 Hz, H-3α), 2.76 (dd, 1H, *J* = 14.8, 7.4 Hz, H-6α), 3.01 (s, 1H, H-17), 3.19 (dd, 1H, *J* = 10.2, 7.1 Hz, H-1), 3.27–3.33 (m, 1H, H-16), 3.29 (s, 3H, CH_3_O-1), 3.30 (s, 3H, CH_3_O-16), 3.40 (s, 3H, CH_3_O-14), 3.43 (d, 1H, *J* = 4.7 Hz, H-14), 3.55 (br.s, 1H, OH), 3.61 (d, 1H, *J* = 11.4 Hz, H-19α), 5.22 (br.s, 2H, NH_2_), 7.34 (s, 1H, H-12′), 7.60 (d, 2H, *J* = 8.6 Hz, H-3″,5″), 7.91 (d, 2H, *J* = 8.6 Hz, H-2″,6″), 8.15 (dd, 1H, *J* = 8.9, 2.2 Hz, H-4′), 8.63 (d, 1H, *J* = 2.2, H-6′), 8.80 (d, 1H, *J* = 8.8, H-3′), 11.14 (s, 1H, NH); ^13^C NMR (101 MHz, CDCl_3_, δ, ppm): 13.4 (CH_3_-22), 24.0 (C-6), 25.5 (CH_3_C(O)), 26.0 (C-12), 26.6 (C-2), 31.7 (C-3), 36.1 (C-13), 44.7 (C-15), 47.4 (C-7), 48.4 (C-5), 48.8 (C-21), 49.6 (C-10), 50.8 (C-11), 55.3 (C-19), 56.0 (CH_3_O-16), 56.4 (CH_3_O-1), 57.8 (CH_3_O-14), 61.4 (C-17), 75.4 (C-8), 78.4 (C-9), 82.7 (C-16), 83.9 (C-1), 85.0 (C-4), 89.9 (C-14), 102.9 (C-12′), 115.8 (C-1′), 120.1 (C-3′); 124.8 (C-4″), 128.5 (C-2″,6″), 129.8 (C-6′), 131.0 (C-5′), 131.7 (C-3″,5″), 132.6 (C-4′), 136.3 (C-1″), 143.1 (C-2′), 163.4 (C-7′); 164.5 (C-11′), 164.8 (C-9′), 166.9 (C=O), 169.0 (CH3C(O)NH); IR (KBr, ν, cm^−1^): 3500, 3388 (OH, NH), 1703 (C=O), 1684 (C=O amide), 1539, 1591 (C=N); HR-MS (ESI), *m*/*z* (I*_rel_*, %): 831 (1), 800 (3), 555 (2), 479 (2), 456 (3), 440 (7), 428 (23), 384 (25), 376 (100), 366 (10), 340 (5), 43 (6); calcd. for C_42_H_50_BrN_5_O_8_: 831.2661; found, [M − OCH_3_]^+^
*m*/*z*: 800.2666.

#### 3.2.7. 4β-{2′-Acetylamino-5′-[9′-(methyl)-11′-(phenyl)pyrimidine-7′-yl]benzoate}-1α,14α,16β-trimethoxy-20-ethylaconitane-8,9-diol (**20**)

Yield 92% (a), 71% (b), 62% (c); yellow amorphous powder; mp 141.3 °C (decomp.); ^1^H NMR (500 MHz, CDCl_3_, δ, ppm): 1.12 (t, 3H, *J* = 7.1 Hz, CH_3_-22), 1.68 (dd, 1H, *J* = 14.8, 8.3 Hz, H-6β), 1.88 (br.t, 1H, *J* = 12.5 Hz, H-3β), 1.97 (m, 1H, H-12β), 2.00 (m, 1H, H-15β), 2.10 (dd, 1H, *J* = 12.4, 4.4 Hz, H-10), 2.15–2.22 (m, 2H, H-2β, H-7), 2.25 (s, 3H, CH_3_C(O)), 2.23–2.33 (m, 2H, H-2α, H-13), 2.34–2.44 (m, 2H, H-15α, H-5), 2.46–2.54 (3H, m, H-12α, H-21α, H-19β), 2.54–2.63 (m, 2H, H-21β, OH), 2.69 (dd, 1H, *J* = 12.5, 2.4 Hz, H-3α), 2.76 (dd, 1H, *J* = 14.8, 7.4 Hz, H-6α), 2.83 (s, 3H, CH_3_-pyrimidine), 3.02 (s, 1H, H-17), 3.20 (dd, 1H, *J* = 10.2, 7.1 Hz, H-1), 3.25–3.34 (m, 1H, H-16), 3.29 (s, 3H, CH_3_O-1), 3.30 (s, 3H, CH_3_O-16), 3.40 (s, 3H, CH_3_O-14), 3.43 (d, 1H, *J* = 4.7 Hz, H-14), 3.52 (br.s, 1H, OH), 3.59 (d, 1H, *J* = 11.4 Hz, H-19α), 7.49 (m, 3H, H-3″,4″,5″), 7.81 (s, 1H, H-12′), 8.05–8.15 (m, 2H, H-2″,6″), 8.23 (dd, 1H, *J* = 8.9, 2.2 Hz, H-4′), 8.75 (d, 1H, *J* = 2.2, H-6′), 8.83 (d, 1H, *J* = 8.9, H-3′), 11.12 (s, 1H, NH); ^13^C NMR (125 MHz, CDCl_3_, δ, ppm): 13.4 (CH_3_-22), 24.0 (C-6), 25.5 (CH_3_C(O)), 26.0 (C-12), 26.3 (CH_3_ pyrimidine), 26.6 (C-2), 31.7 (C-3), 36.1 (C-13), 44.7 (C-15), 47.4 (C-7), 48.4 (C-5), 48.8 (C-21), 49.6 (C-10), 50.8 (C-11), 55.3 (C-19), 56.0 (CH_3_O-16), 56.4 (CH_3_O-1), 57.8 (CH_3_O-14), 61.4 (C-17), 75.4 (C-8), 78.4 (C-9), 82.7 (C-16), 83.9 (C-1), 85.0 (C-4), 89.9 (C-14), 109.0 (C-12′), 116.0 (C-1′), 120.3 (C-3′); 127.1 (C-2″,6″), 128.7 (C-3″,5″), 130.1 (C-6′), 130.4 (C-4″),131.1 (C-5′), 132.6 (C-4′), 137.3 (C-1″), 143.1 (C-2′), 162.9 (C-7′); 164.7 (C-11′), 166.9 (C=O), 168.4 (C-9′), 168.9 (CH_3_C(O)NH); IR (KBr, ν, cm^−1^): 3400 (OH, NH), 1704 (C=O), 1685 (C=O amide), 1537, 1587 (C=N); HR-MS (ESI), *m*/*z* (I*_rel_*, %): 752 (1), 721 (2), 528 (2), 441 (2), 412 (12), 410 (35), 376 (100), 84 (2); calcd. for C_43_H_52_N_4_O_8_: 752.3598; found, [M − OCH_3_]^+^
*m*/*z*: 721.3612.

#### 3.2.8. 4β-{2′-Acetylamino-5′-[9′-(amino)-11′-(phenyl)pyrimidine-7′-yl]benzoate}-1α,14α,16β-trimethoxy-20-ethylaconitane-8,9-diol (**21**)

Yield 90% (a), 75% (b), 71% (c); yellow amorphous powder; mp 169.0 °C (decomp.); ^1^H NMR (400 MHz, CDCl_3_, δ, ppm): 1.11 (t, 3H, *J* = 7.1 Hz, CH_3_-22), 1.66 (dd, 1H, *J* = 14.8, 8.3 Hz, H-6β), 1.87 (br.t, 1H, *J* = 12.5 Hz, H-3β), 1.96 (m, 1H, H-12β), 2.00 (m, 1H, H-15β), 2.09 (dd, 1H, *J* = 12.4, 4.4 Hz, H-10), 2.14–2.22 (m, 2H, H-2β, H-7), 2.24 (s, 3H, CH_3_C(O)), 2.23–2.32 (m, 2H, H-2α, H-13), 2.33–2.43 (m, 2H, H-15α, H-5), 2.44–2.53 (3H, m, H-12α, H-21α, H-19β), 2.54–2.61 (m, 2H, H-21β, OH), 2.66 (dd, 1H, *J* = 12.5, 2.4 Hz, H-3α), 2.75 (dd, 1H, *J* = 14.8, 7.4 Hz, H-6α), 3.00 (s, 1H, H-17), 3.18 (dd, 1H, *J* = 10.2, 7.1 Hz, H-1), 3.26–3.32 (m, 1H, H-16), 3.28 (s, 3H, CH_3_O-1), 3.29 (s, 3H, CH_3_O-16), 3.39 (s, 3H, CH_3_O-14), 3.42 (d, 1H, *J* = 4.7 Hz, H-14), 3.55 (br.s, 1H, OH), 3.58 (d, 1H, *J* = 11.4 Hz, H-19α), 5.24 (s, 2H, NH_2_), 7.39 (s, 1H, H-12′), 7.47 (m, 3H, H-3″,4″,5″), 8.03 (m, 2H, H-2″,6″), 8.15 (dd, 1H, *J* = 8.9, 2.2 Hz, H-4′), 8.66 (d, 1H, *J* = 2.2, H-6′), 8.80 (d, 1H, *J* = 8.9, H-3′), 11.14 (s, 1H, NH); ^13^C NMR (125 MHz, CDCl_3_, δ, ppm): 13.4 (CH_3_-22), 24.1 (C-6), 25.5 (CH_3_C(O)), 26.2 (C-12), 26.7 (C-2), 31.8 (C-3), 36.3 (C-13), 44.7 (C-15), 47.6 (C-7), 48.4 (C-5), 48.8 (C-21), 49.8 (C-10), 51.0 (C-11), 55.5 (C-19), 56.0 (CH_3_O-16), 56.4 (CH_3_O-1), 57.8 (CH_3_O-14), 61.3 (C-17), 75.5 (C-8), 78.5 (C-9), 82.8 (C-16), 84.0 (C-1), 85.1 (C-4), 90.1 (C-14), 103.4 (C-12′), 116.0 (C-1′), 120.2 (C-3′); 127.0 (C-2″,6″), 128.6 (C-3″,5″), 129.9 (C-6′), 130.4 (C-4″), 131.4 (C-5′), 132.6 (C-4′), 137.6 (C-1″), 143.1 (C-2′), 163.5 (C-7′); 164.4 (C-11′), 166.2 (C-9′), 167.0 (C=O), 169.0 (CH_3_C(O)NH); IR (KBr, ν, cm^−1^): 3400, 3369 (OH, NH), 1703 (C=O), 1683 (C=O amide), 1542, 1589 (C=N); HR-MS (ESI), *m*/*z* (I*_rel_*, %): 753 (1), 722 (4), 528 (10), 513 (12), 410 (17), 407 (12), 392 (30), 376 (100), 58 (5); calcd. for C_42_H_51_N_5_O_8_: 753.3558; found, [M − OCH_3_]^+^
*m*/*z*: 722.3561.

### 3.3. Biological Evaluation

#### 3.3.1. Pharmacology

##### Animals

All studies were carried out on non-breeding CD-1 albino mice (male) weighting 20–25 g, 8 animals in each group (SPF-vivarium of the Institute of Cytology and Genetics of the Siberian Branch of the Russian Academy of Sciences). Mice were maintained at 22–25 °C on a 12 h light-dark cycle with food and water available ad libitum. All work with animals was performed in strict accordance with the legislation of the Russian Federation, the regulations of the European Convention for the Protection of Vertebrate Animals Used for Experimental and Other Scientific Purposes (1986), and the requirements and recommendations of the Guide for the Care and Use of Laboratory Animals and was approved by the Ethic Committee of the N.N. Vorozhtsov Institute of Organic Chemistry SB RAS (protocol No. 2/2016 from 15.02.2016, protocol No. 8/2017 from 09.11.2017).

##### Analgesic Tests

Agents were dissolved in saline containing 0.5% Tween 80 just before use and were administered per os, 1 h before testing. Saline was administered per os in blank mice (control group), 1 h before testing. Analgesic activity of test agents was assessed using acetic acid-induced writhing test and hot plate test. The same animals were used in both tests. The first was the “acetic acid writhing” test and then animals were exposed to the “hot plate” test. In the acetic acid-induced writhing test, the pain reaction was determined by the number of abdominal convulsions, recorded from the 5th to the 8th min following the acetic acid injection (0.75%, 0.1 mL/10 g body weight) [51]. The percentage of pain reaction inhibition was calculated according to the following equation: % inhibition = 100 × (A − B)/A, where A is the mean number of writhes in the control group, and B is the mean number of writhes in the test group. In the hot plate test, animals were placed individually on a metallic plate (VWR Hotplate/Stirrer 725-HPS, Radnor, PA, USA) warmed to 54 ± 0.5 °C until either licking of the hind paw or jumping [56]. This time of pain response was recorded by a stopwatch and the animal was immediately taken away from the plate and put back into the cage. Statistical analysis was conducted in Statistica 7.0 program using the Mann–Whitney U Test to assess the significance (*p* < 0.05) of differences. The data are presented in the format: mean ± standard error of the mean (SE).

### 3.4. Cell Culture and Determination of Cytotoxicity

The human cancer cells of HepG2 (hepatocellular carcinoma), Hek293 (fetal kidney), U937 (histiocytic lymphoma), and Hep2 (laryngeal carcinoma) were used in this study. The cells were cultured in the IMDM (Sigma-Aldrich, St. Louis, MO, USA) (for Hep2, HepG2, Hek293), and RPMI-1640 (HyClone, Germany) (U937) medium that contained 10% embryonic calf serum (HyClone), in a CO_2_ incubator at 37 °C. Cells were placed on 96-well microliter plates and cultivated at 37 °C in 5% CO_2_/95% air incubated for 24 h then the tested compounds dissolved in DMSO (concentration <1%) were added to the cellular culture at the required concentrations (10–100 µM) and incubated for 48 h. Two wells were used for each concentration. The cells which were incubated without the compounds were used as a control. The cell viability was assessed through the method of double staining with the fluorescent dyes Hoechst 33342 and propidium iodide (PI). Cells were stained with Hoechst 33342 (Sigma-Aldrich) and propidium iodide (Invitrogen) for 30 min at 37 °C. The tumor cell viability was analyzed using the “In Cell Investigator” program on an IN Cell Analyzer 2200 in an automatic mode to determine living, dead and apoptotic cells in the entire population. The results (Figure 1A–H and Figure 2A–H) are presented as the percentage of cells from two independent experiments.

### 3.5. Molecular Docking Study

Molecular modeling was carried out in the CCDC Hermes 1.10.5 visualization environment using applications from the CSD Discovery 2020 package [57]. Three-dimensional structures of the derivatives were obtained empirically in the Conformer Generator [58] application. For the calculations, the XRD model of voltage-gated sodium (Nav1.7) channel with PDB ID 5EK0 [59] (resolution 3,53 Å) from the Protein Data Bank was chosen. To model a possible mechanism of inhibition of selected target, molecular docking of new compounds was performed at the binding site of Nav1.7 antagonist using GOLD [60]. The search area for docking was selected according to the size of antagonist GX-936. Docking was performed in comparison with the GX-936 molecule. The three-dimensional structures of antagonist were obtained in the PubChem database and prepared in the Conformer Generator application. Non-covalent interactions of compounds in the binding site were visualized using BIOVIA Discovery Studio Visualizer [61].

## 4. Conclusions

An expedient one-pot sequential, four component synthesis of substituted lappaconitine–pyrimidine hybrids is described. The PdCl_2_-Ad_2_PBn·HBr catalytic system was successfully applied in the multicomponent carbonylative cross-coupling-cyclocondensation affording diterpenoid-pyrimidine hybrids in good yields of up to 71%. The protocol provides a mild reaction condition, high yield of the product, and operational simplicity to assemble complex structural entity in a single operation. We found, that the new type of hybrid compounds exhibited pronounced antinociceptive activity in in vivo tests: hot plate and acetic acid-induced writhing tests in the dose of 5 and 1 mg/kg by oral or intraperitoneal administration. The activity of compounds **10**, **12**, **13**, and **15** was comparable to those of the standard drug Diclofenac sodium, used in the dose of 10 mg/kg and lappaconitine in the dose of 5 mg/kg. The leading compound **12** showed moderate oral toxicity (LD_50_ value was more than 600 mg/kg, 20 times less toxic than the parent compound lappaconitine **1**) and low cytotoxicity in in vitro tests. In addition, the proposed transformations due to their simplicity and effectiveness, will find use in the development of derivatives with the lappaconitine core, which could be used as scaffolds toward accessing other libraries of bioactive compounds.

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
