# Peer review of "Hybrides of Alkaloid Lappaconitine with Pyrimidine Motif on the Anthranilic Acid Moiety: Design, Synthesis, and Investigation of Antinociceptive Potency"

_molecules, 2020, doi:10.3390/molecules25235578_

Round 1
Reviewer 1 Report
The authors reported convenient and efficient routes to pyrimidine based lappaconitine derivatives through both Sonogashira coupling/cyclocondensation and palladium-catalyzed carbonylative Sonogashira coupling/cyclocondensation sequences. The authors have also demonstrated an expedient one-pot sequential four-component synthesis of substituted lappaconitine pyrimidine could be additional merit to this method. The compounds (10,12,13 and 15) possesses high activity towards antinociceptive activity in vivo tests when the substituent is present at the 2 and 6 positions of the pyrimidine nucleus. The presentation needs to be improved particularly at the Schemes 1-4 before publication. It is hard to follow as the numbers are repeating twice. The following corrections are recommended. Overall, it is a good study and therefore recommended for publication in Molecules after minor revision.
Additional comments:
- Line 106; Indicate the substitution at Ar in 3,4 as like 3: Ar = 4-FC6H4 and 4: Ar = 4-CH3OC6H4. For alkynyl ketones 5,6; remove 3,4 as it is not necessary and confusing. Please follow as represented above for all molecules in Scheme 1-3. For amidines 9,10; include R = Me and R = NH2 along with its salts, respectively. For pyrimidines 7,8; remove 9,10 as it is not necessary and confusing.
- Line 119; remove 15,16 from 15: R = CH3 and 16: R = NH2 as it is repeating in Scheme 2. Also, please indicate the exact base (Na2CO3?) for the conditions (b) and followed throughout the manuscript. Just mentioning base is not enough.
- Line 161; remove 20,21 from 20: R = CH3 and 21: R = NH2 as it is repeating in Scheme 3.
- Line 172; remove hyphens from compound 19 and 22 in Scheme 4.
Author Response
Thank you very much for all the comments. All Schemes were modified.
Authors were very grateful for the valuable remarks from Referee 1. We made important corrections and additions to the manuscript, which were necessary for the better presentation of our scientific material.
Sincerely,
Elvira Shults

Reviewer 2 Report
In this manuscript, Shults and collaborators report the synthesis of some hybrids of lappaconitine pyrimidine and the evaluation of their biological activities.
I do not recommend this article for publication on Molecules due to the lack of novelty in the chemistry section of the manuscript. Indeed, both the synthetic approach and the synthesized products are not new.
In fact, the authors claim for improved synthetic procedures by avoiding the purification of the intermediate in one case (compounds 15 and 16) and by setting up a sequential one-pot procedure that involves a three-component reaction and a sequential one-pot step (compounds 20- 21) in the other case. In my view, both the expedients have not implemented innovative chemistry to access these products.
Other issues:
1) The strategy to access products 20 and 21 cannot be conceived as a four-component reaction.
2) The schemes are not presented clearly: for example, yields are missing; the comparison between the already reported procedure and the new procedure is not clearly highlighted etc.
3) The English needs a significant revision: puzzling expressions like "around theses" (line 84, page 2), "mine problem" (line 123, page 4), should be amended.
4) The title is also rather disconcerting: "Optimizing the Lappaconitine alkaloid ..." doesn't seem to make any sense.
Author Response
Thank you very much. We added several comments in the text.
Other issues:
1) The strategy to access products 20 and 21 cannot be conceived as a four-component reaction.
Thank you very much. This was an error, it was correct.
2) The schemes are not presented clearly: for example, yields are missing; the comparison between the already reported procedure and the new procedure is not clearly highlighted etc.
We have made some correction in the Scheme (added yields and reagents) and also corrected the text.
3) The title is also rather disconcerting: "Optimizing the Lappaconitine alkaloid ..." doesn't seem to make any sense.
In the Title we want to highlight the fact that the specified modification made it possible to improve the structure in terms of reducing toxicity and improving activity. We also made correction in the text.
Authors were very grateful for the valuable remarks from Referee 2. We made important corrections and additions to the manuscript, which were necessary for the better presentation of our scientific material.
Thank you very much for all the comments.

Reviewer 3 Report
This manuscript deals with the functionalisation of an alkaloid natural product by a multicmponent reaction leading to a diaryl pyrimidinyl substituent. It is based on earlier work on Sonogashira functionalisation of this alkaloid. A small library of compounds is prepared and biological properties investigated.
From the viewpoint of organic synthesis this could be considered as routine, no really new chemistry is employed here.
The rationale of these compounds is not that well explained, what do we expect of these hybrids, why exactly this combination and why in this way ?
The diarylpyrimidine moiety is connected through an ester bond, I wonder how stable that would be if we would apply such a compound in vivo. Metabolism studies could be useful.
Schemes should be in uniform style (bond length, size, etc.) throughout
The mechanism of page 5 is not needed, this is well known, due reference can be given instead.
Small corrections to the English are needed, to give one example p. 4 line 123 main not mine, p. 9 thiadiazole
Author Response
This manuscript deals with the functionalisation of an alkaloid natural product by a multicmponent reaction leading to a diaryl pyrimidinyl substituent. It is based on earlier work on Sonogashira functionalisation of this alkaloid. A small library of compounds is prepared and biological properties investigated.
From the viewpoint of organic synthesis this could be considered as routine, no really new chemistry is employed here.
The rationale of these compounds is not that well explained, what do we expect of these hybrids, why exactly this combination and why in this way?
Thank you very much. We have added any comment in the text.
The diarylpyrimidine moiety is connected through an ester bond, I wonder how stable that would be if we would apply such a compound in vivo. Metabolism studies could be useful.
Thank you very much for this remark. We made several additions in the Introduction part deals with the metabolism of the parent compound lappaconitine (about the metabolic stability of the ester bond).
Line 76: The metabolism in animals and humans have been clearly studied for lappaconitine. The results obtained provided basic information to help better understand the pharmacological and toxicological activities of LAP in human indicated that the ester bond of lappaconitine is stable [22-24]. 2ʹ-N-deacetyllappaconitine and 16-O-demethyllappaconitine are the main metabolic products of lappaconitine. The findings indicated that lappaconitine was widely metabolized: N-deethylation, O-demethylation, hydrolysis and hydroxylation were main metabolic pathways, but that dehydroxylation and ester-exchange reactions were not found [24].
[22] Xie, F.; Wang, H.; Shu, H.; Li, J.; Jiang, J.; Chang, J.; Hsieh, Y. Separation and characterization of the metabolic products of lappaconitine in rat urine by high-performance liquid chromatography. J. Chromatogr. B: Biomed. Sci. Appl. 1990, 526, 109–118.
[23] Xie, F.; Wang, H.C.; Li, J.H.; Shu, H.L.; Jiang, J.R.; Chang, J.P.; Hsieh, Y.Y. Studies on the metabolism of lappaconitine in humans. Identification of the metabolites. Biomed. Chromatogr. 1990, 4, 43-48.
[24] Yang, S.; Zhang, H.; Beier, R.C.; Sun, F.; Cao, H.; Zhen, j.; Wang, Z.; Zhang S. Comparative metabolism of Lappaconitine in rat and human livermicrosomes and in vivo of rat using ultra high-performance liquidchromatography–quadrupole/time-of-flight mass spectrometry. J. Pharm. Biomed. Anal. 2015, 110, 1-11.
Schemes should be in uniform style (bond length, size, etc.) throughout
The correction was made on the Schemes according to the remark of the Reviewer
The mechanism of page 5 is not needed, this is well known, due reference can be given instead.
The reference was added. By giving this scheme we only wanted to show the usefulness of catalytic systems based on PdCl2 (PdCl2 – Ad2PBn·HBr and PdCl2 – Ph3P) for the selectivity of this cross-coupling towards the formation of alkynyl ketone, and not cross-coupling product 22. We have correct the text.
Small corrections to the English are needed, to give one example p. 4 line 123 main not mine, p. 9 thiadiazole
We have correct these errors.
Authors were very grateful for the valuable remarks from Referee 3. We made important corrections and additions to the manuscript, which were necessary for the better presentation of our scientific material. Thank you very much for all the comments.
Sincerely,
Elvira Shults

Round 2
Reviewer 2 Report
This is a revised version of the manuscript I reviewed earlier.
Although the article has been slightly revised, my concern about the novelty of the synthetic part still remains. Indeed, the Authors, through known synthetic protocols, addressed the synthesis of lappaconitine derivatives already reported.
The studies concerning the evaluation of the biological activities could still be interesting, but, in the article, it was not demonstrated that this synthetic approach allows the synthesis of new (optimized?) structures for which interesting results, in terms of biological activities, were expected. On the other hand, biological tests were carried out on structures for which synthetic protocols already existed.
In the title, the authors claim a generic "optimization of the lappaconitine alkaloid with ...": what they refer to still remains obscure. To the synthesis? to the structure? "optimization of <<compound name>>" does not make any sense.
English language still need to be revised.